A hierarchical model of daily stream temperature using air-water temperature synchronization, autocorrelation, and time lags

Letcher Benjamin H. 1 bletcher@usgs.gov
Hocking Daniel J. 1
O’Neil Kyle 1
Whiteley Andrew R. 2
Nislow Keith H. 3
O’Donnell Matthew J. 1
1 S.O. Conte Anadromous Fish Research Center, US Geological Survey/Leetown Science Center , Turners Falls , USA
2 Department of Environmental Conservation, University of Massachusetts , Amherst , USA
3 Northern Research Station, USDA Forest Service, University of Massachusetts , Amherst, MA , USA
Stanford Jack
Electronic publication date: 2016 Feb 29
Publication date: 2016
Volume: 4
Electronic Location ID: e1727
Received 2015 Dec 9; Accepted 2016 Feb 2
Copyright year: 2016
License: This is an open access article, free of all copyright, made available under the Creative Commons Public Domain Dedication. This work may be freely reproduced, distributed, transmitted, modified, built upon, or otherwise used by anyone for any lawful purpose.
License URL: https://creativecommons.org/publicdomain/zero/1.0/

Keywords: Stream temperature, Ecology, Air temperature, Statistical model, Climate change

Funding: USFWS North Atlantic Conservation Cooperative and the USGS Northeastern Climate Science Center Funding was provided by the USFWS North Atlantic Conservation Cooperative and the USGS Northeastern Climate Science Center. The funders had no role in study design, data collection and analysis, decision to publish, or preparation of the manuscript.

==============================
Water temperature is a primary driver of stream ecosystems and commonly forms the basis of stream classifications. Robust models of stream temperature are critical as the climate changes, but estimating daily stream temperature poses several important challenges. We developed a statistical model that accounts for many challenges that can make stream temperature estimation difficult. Our model identifies the yearly period when air and water temperature are synchronized, accommodates hysteresis, incorporates time lags, deals with missing data and autocorrelation and can include external drivers. In a small stream network, the model performed well (RMSE = 0.59°C), identified a clear warming trend (0.63 °C decade−1) and a widening of the synchronized period (29 d decade−1). We also carefully evaluated how missing data influenced predictions. Missing data within a year had a small effect on performance (∼0.05% average drop in RMSE with 10% fewer days with data). Missing all data for a year decreased performance (∼0.6 °C jump in RMSE), but this decrease was moderated when data were available from other streams in the network.

Introduction

Accurate stream temperature predictions are increasingly important as human impacts on streams and on the climate accelerate stream temperature change (Kaushal et al., 2010; Rice & Jastram, 2015). Human activities influence stream temperatures directly via increased water withdrawals, altered channel engineering and dam operation (Poole & Berman, 2001) and indirectly by altering landscape features (e.g., riparian cover) and by affecting air temperatures at broad spatial scales via climate change (Hayhoe et al., 2007; Huntington et al., 2009). Understanding how stream temperatures are changing over time and space and the ability to forecast future temperatures are important because stream temperatures directly influence stream ecosystems (Quinn et al., 1994; Wenger et al., 2011) and because regulatory agencies commonly use stream temperature as a metric for managing streams and their watersheds (e.g., Beauchene et al., 2014). Altered stream temperatures are likely to have profound effects on the abundance and distribution of stream biota (Isaak & Rieman, 2013; Eby et al., 2014), especially coldwater, ectothermic species because many physiological and demographic rates are temperature-dependent (Fry, 1971; Elliott & Elliott, 2010; Letcher et al., 2015).

The general importance of stream temperature has prompted the development of a number of models for stream temperature (e.g., Mohseni, Stefan & Erickson, 1998; Caissie, El-Jabi & Satish, 2001; Hague & Patterson, 2014; Sun et al., 2014; Li et al., 2014). Stream temperature models vary along several important gradients, including model type (physical-statistical), temporal resolution (daily-yearly) and spatial resolution (local-broad spatial coverage). As with all models of complex systems, tradeoffs among these gradients usually limit models to highly-detailed, local models (Brown, 1969; Kim & Chapra, 1997; Younus, Hondzo & Engel, 2000) or simple, general models (e.g., Crisp & Howson, 1982). The detailed, local models typically produce good accuracy (RMSE ∼1.0 °C) but may not predict temperatures well outside of the local area, while the simple models generate moderate to poor accuracy (RMSE ∼1.5–3.0 °C) across a broad spatial range. Models that aggregate over longer time intervals generally perform better (Stefan & Preud’homme, 1993; Pilgrim, Fang & Stefan, 1998; Webb, Clack & Walling, 2003; Morrill, Bales & Conklin, 2005), but even hourly models can perform well (Kanno, Vokoun & Letcher, 2014). A careful consideration of six key temperature modeling issues may provide the basis for the development of effective daily stream temperature models of medium complexity.

First, the relationship between air temperature and stream temperature is non-linear at high and low air temperatures (Mohseni, Stefan & Erickson, 1998), but for different reasons. At high air temperatures, evaporative cooling slows warming of stream water, while at low air temperatures, air temperatures can dip well below the water temperature freezing limit (Caissie, 2006; Webb et al., 2008). Air and water temperatures are no longer synchronized when air temperatures are near and below 0 °C, which can generate a poor relationship between air and stream temperatures and heterogeneity of variance across temperatures. Many simple statistical models use a non-linear model to describe the relationship between air and stream temperature (Mohseni, Stefan & Erickson, 1998; Webb, Clack & Walling, 2003; Kanno, Vokoun & Letcher, 2013). Others use a linear model and limit analysis to the summer (Hilderbrand, Kashiwagi & Prochaska, 2014) or to the ice-free period of the year (Stefan & Preud’homme, 1993; Erickson & Stefan, 2000), in an attempt to avoid the non-linear portions of the air-water temperature relationship. Time series (Caissie, El-Jabi & St-Hilaire, 1998; Caissie, El-Jabi & Satish, 2001; Benyahya et al., 2007) or non-parametric models (Benyahya et al., 2008; Li et al., 2014) of stream temperature trends over time that include air temperature as a predictor as well as local, physical models (e.g., Sinokrot & Stefan, 1993) can accommodate the non-linearity.

Second, accuracy can be improved when models account for hysteresis, a different relationship between air and water temperature in the spring (rising temperatures) vs. the fall (falling temperatures) (Mohseni, Stefan & Erickson, 1998; Caissie, El-Jabi & Satish, 2001; Webb, Clack & Walling, 2003). Seasonal hysteresis is often caused by influx of cool snow melt or rain water in the spring (Lisi et al., 2015) which depresses spring stream temperature/air temperature relationships relative to fall stream temperature/air temperature relationships (Webb & Nobilis, 1997). Mohseni, Stefan & Erickson (1998) observed that 43% of their study streams exhibited hysteresis; they addressed hysteresis by fitting separate non-linear curves to the rising and falling seasonal temperatures. Time series models with non-symmetric seasonal functions account for hysteresis by default (e.g., Li et al., 2014).

Third, due to thermal inertia, stream temperature does not respond instantaneously to changes in air temperature. Including lags in air temperature effects can improve estimates for models with short time scales (Benyahya et al., 2008; Webb, Stewardson & Koster, 2010). The effects of time lags increase with stream depth (Stefan & Preud’homme, 1993) and stream flow (Smith & Lavis, 1975; Webb, Clack & Walling, 2003). Time lags are a key component of time series modeling (Shumway & Stoffer, 2006).

Fourth, while the amount of stream temperature data available worldwide is increasing very rapidly (Webb et al., 2008), many sites have incomplete data. Very few study regions have a complete matrix of sample sites and years: data may be missing for an entire year at a site or may be incomplete within a year. Incomplete within-year data will have variable effects on estimation depending on the extent and timing of the missing data. Effects of missing data will also depend on model type. For simple linear models, within-year missing data may not have a large effect on estimation because of the linear relationship between stream and air temperature. For non-linear models, missing data could have dramatic effects on estimation as missing data fail to ‘anchor’ the curve. Other modeling approaches, such as time series models , machine learning models (DeWeber & Wagner, 2014), and models with varying coefficients (Li et al., 2014) may be less sensitive to missing data. In general, hierarchical models with random effects across space (sites, stream networks or regions) and time (months, seasons, or years) can accommodate missing data as they ‘borrow information’ across units (Wagner, Hayes & Bremigan, 2006; Gelman & Hill, 2006).

Fifth, spatial and temporal autocorrelation can cause estimation problems (Caissie, 2006; Benyahya et al., 2007; Hague & Patterson, 2014). Autocorrelation occurs when data points in space or time are not independent, i.e., close points are similar or dissimilar to each other simply because they are close. For example, downstream temperatures can be similar to upstream temperatures because water flows downstream or today’s temperature can be similar to yesterday’s temperature due to the combination of high heat capacity of water, low density and heat transfer from air, and conduction of heat from surrounding environment (i.e., thermal inertia) (Caissie, El-Jabi & St-Hilaire, 1998; Isaak et al., 2014). This is a very common issue in estimation and a variety of time series models can accommodate temporal autocorrelation (Shumway & Stoffer, 2006) and some newer approaches are now available to deal with spatial autocorrelation (Peterson & Ver Hoef, 2010; Rushworth et al., 2015).

Finally, air temperature is not the only important predictor of stream temperature (Webb, 1996; Caissie, 2006). Many regression-based models have evaluated effects of landscape and environmental drivers on stream temperatures (Hawkins et al., 1997; Isaak & Hubert, 2001; Hill, Hawkins & Carlisle, 2013). Important landscape drivers typically include topography, riparian cover, impervious surface, and stream depth (Poole & Berman, 2001) and environmental drivers often include stream flow, snow melt, groundwater input, and humidity (Taylor et al., 2013; Lisi et al., 2015; Snyder, Hitt & Young, 2015). It is generally straightforward to incorporate external drivers beyond air temperature into most classes of stream temperature models (Hague & Patterson, 2014).

Here, we develop a model for mean daily stream temperature that improves accuracy of statistical models by addressing most of the issues listed above. To avoid fitting a relationship between stream and air temperature when there is none (e.g., winter), we develop a metric that limits estimation to the days of the year that daily stream temperature and daily air temperature are synchronized (roughly spring to fall). This metric is flexible among years and sites. To address hysteresis, we estimate a non-symmetrical trend across the synchronized days with a hierarchical structure to accommodate missing data. We also add an autoregressive term to the model to deal with temporal autocorrelation and we estimate spatial covariance to accommodate spatial autocorrelation. Because data presented here are spatially constrained to four sites in a small network, we do not include landscape variables in the model. The two environmental drivers in the model are air temperature and stream flow. In addition to presenting the model, we analyze trends in estimates over time and conduct a detailed missing observations analysis.

Methods

Study area

The study site was located in western Massachusetts, USA (42°25′N; 72°39′W, Fig. 1) and consisted of a third-order mainstem (West Brook, WB) and three second-order tributaries (Open large, OL; Open small, OS; Isolated large, IL) (Fig. 1 and Table 1). A dense canopy of mixed hardwood with some hemlocks provides cover throughout the watershed. Watershed area above our study area is 11.8 km2 and landuse in the area is limited residential with some farming. The study is in an area of low to moderate relief, with altitude ranging from to 100 m to 250 m (Fig. 1). Average stream width of the WB is 4.5 m and is between 1–3 m for the tributaries. Water is stored in two of the streams; a drinking water reservoir is upstream of the WB, and a large beaver dam complex is above OS (Fig. 1). OL and IL were free-flowing during the course of the study.

Figure 1 Map of the study area.

Dots indicate locations of temperature loggers and shading represents elevation (range approximately 100 m–250 m).

Table 1 Landscape variables for the four study streams (Fig. 1).

Stream	Abbrev.	Watershed area (km2)	Impounded area (%)	Watershed forested (%)	Watershed developed (%)	Agriculture (%)	Mean elevation (m)	Mean slope (%)	
West brook	WB	11.8	2.62	92.0	2.9	3.0	164.8	13.02	
Open large	OL	2.5	0.75	88.6	2.8	7.7	226.1	14.75	
Open small	OS	1.0	2.94	99.1	0.0	0.0	211.0	16.46	
Isolated large	IL	1.4	0.58	95.5	1.6	2.9	232.9	16.15	

We deployed temperature loggers (±0.1 C; Optic StowAway and Hobo Pro V2, Onset Computer Corporation, Pocasset, MA, USA, and ±0.05 C; Barologger, Solinst Canada Ltd., Georgetown, ON) in four permanently watered sections of the study area (Fig. 1). Stream temperature data are available at http://db.ecosheds.org. All loggers recorded data every two hours throughout the year. The logger in the WB was deployed 1998–2013 and the loggers in the tributaries were out from 2002 to 2013. We do not have continuous air temperature measurements from 2002 to 2013 (Table 1), so we used mean daily air temperature estimates for our study area from Daymet (http://daymet.ornl.gov/). For the years that we do have West Brook air temperature data (2008–2013), the relationship between West Brook and Daymet air temperatures was strong (p-value < 10−16, r2 = 0.91), suggesting that Daymet air temperatures are a good data source for the study site. Stream flow was estimated using a flow extension model (Nielsen, 1999) based on data from a nearby (∼10 km) USGS stream gage (Mill River, Northampton, MA, USA). See Xu, Letcher & Nislow (2010) for details.

Statistical analysis

Descriptive statistics

As a coarse comparison of daily water temperatures, we calculated correlations among sites. We also explored patterns in water temperature over time and among sites by comparing cumulative residuals from a spline fit to all the data (function gam() in R, Fig. 2). We calculated residuals for each water temperature data point and then developed empirical cumulative curves over days of the year for each year and site combination.

Figure 2 Water temperature data (daily means) from all sites and years overlain by a spline (white line).

Breakpoints

The goal is to develop a robust model for the relationship between mean daily water and mean daily air temperature. A key limitation in developing this relationship is that lower water temperatures in the winter are bounded near 0 °C while air temperatures are not. This means that water and air temperatures can become decoupled when air temperatures are cold resulting in only a weak relationship, at best, between water and air temperature. In contrast, as air temperatures warm in the spring and before they get too cold in the autumn, daily water and air temperatures can be synchronized (Figs. 3A and 3B), suggesting the possibility of a strong relationship between daily water and air temperature during the synchronized portion of each year.

Figure 3 Examples of raw air (red) and water (black) temperatures from the WB (A, B) and the temperature index (C, D) used to calculate the temperature breakpoints (vertical lines).

Horizontal lines in (C) and (D) are the 99% confidence intervals of the temperature index for day of year 125–275. Vertical axes on (C) and (D) are truncated to −20–20.

The key to the approach is identifying a breakpoint in the spring when daily water and daily air temperature become synchronized and a breakpoint in the autumn when daily temperatures become desynchronized. To identify the synchronization breakpoints we calculated a simple index (waterT−airT)/waterT (waterT > 0), where waterT was mean daily water temperature and airT was mean daily air temperature (Fig. 3). This temperature index (tempIndex) approaches 0 when water and air temperature are similar and is very different from 0 when temperatures diverge (Figs. 3C and 3D). While water and air temperatures are synchronized, tempIndex flattens out (Figs. 3C and 3D), providing the opportunity to identify the beginning and end (breakpoints) of the flat period.

To identify the spring and autumn breakpoints, we used a runs analysis that determined the first (spring) and last (autumn) day of the year that the tempIndex was consistently within the flat period (Figs. 3C and 3D). We established the range of tempIndex values that comprised the flat period by calculating the 99.9% confidence interval (CI) for tempIndex using the middle 150 days of the year (late April to mid-September). The middle 150 days of the year were always within the flat period based on visual observation of tempIndex plots. Separate CI values were calculated for each year and stream. For the breakpoint estimation, we used a moving average for tempIndex with a centered 10-day window to help stabilize tempIndex values near the breakpoints. Temperatures were considered synchronized when 10 consecutive days of the moving average fell within the 99.9% CI. Beginning on day 1 and moving towards day 150, the first time 10 consecutive days were synchronized was used as the spring breakpoint and we moved from the end of the year to day 150 to establish the fall breakpoint. Numbers of days in the synchronized period for each stream and year are shown in Table 2.

Table 2 Number of synchronized days with stream temperature data for each combination of year and site.

Numbers in parentheses represent the percentage of days with missing data within the synchronized period.

	West brook	Open large	Open small	Isolated large	
1999	233	0	0	0	
2000	256	0	0	0	
2001	230	0	0	0	
2002	237	0	81 (58)	0	
2003	191	179 (4)	183 (2)	180	
2004	222	214 (3)	215 (1)	214	
2005	235	203	102 (52)	200	
2006	210	0	83 (59)	214	
2007	0	192	204	192	
2008	121 (45)	198	199	197	
2009	0	243	251	247	
2010	273	245	265	246	
2011	233	205	248	205	
2012	210	234	237	235	
2013	226	218	0	212	

We evaluated trends in fall and spring breakpoints by running three linear models with breakpoint day of the year as the dependent variable and year alone or year + stream or year * stream as independent variables. We estimated AIC to determine the most parsimonious model.

Water temperature model description

With breakpoints established for each year and site, we modeled the relationship between water temperature and air temperature for the synchronized period using a hierarchical linear autoregressive model with a cubic trend across days within a year and covariation among sites. We fit the model using a Bayesian approach.

Observed water temperature (ts,d,y) for each site (s; s1 = WB, s2 = OL, s3 = OS, s4 = IL), day of year (d) and year (y) was assumed to derive from a normal distribution with mean μs,d,y and standard deviation sd (residual model error): (1) ts,d,y∼Nμs,d,y,sd.

We used a non-informative uniform prior [0, 10] for sd. We modeled the mean with a linear trend (ωs,d,y) adjusted by an AR(1) autoregressive coefficient (δs) on the residual error from the previous day: (2) μs,d,y=ωs,d,y+δsts,d−1,y−ωs,d−1,y.

We placed a hierarchical structure on δs: (3) δs∼Nμδs,sdδT−1,1

where site-specific δs were drawn from a truncated normal distribution with mean μδs and standard deviation sdδ. Values for δs were truncated to keep them within the admissible range for a correlation. Priors for the mean and standard deviation were non-informative; μδs ∼ U(−1, 1), and sdδ ∼ U(0, 2) (an upper limit of 2 for sdδ is non-informative for the truncated data).

When observed temperature data were not available for the previous day (beginning of a series or following a break in the series) we modeled the mean without the autoregressive component: (4) μs,d,y=ωs,d,y.

We modeled the linear component with a combination of fixed and random effects: (5) ωs,d,y=α+β1Ts,d,y+β2Ts,d−1,y+β3Ts,d−2,y+β4Fs,d,y+β5Ts,d,y⋅Fs,d,y+β6:8s+β9:11s⋅Ts,d,y+Yy

where α is the overall intercept, the β are the coefficients for the fixed effects (T is mean daily air temperature, F is mean daily stream flow, s is site) and Yy represents random effects among years. Priors for the β1:11 were independent and non-informative, N(0, 100).

Yy represented random effect temporal trends (cubic) across years where: (6) Yy=αy+β12,yDs,y,d+β13,yDs,y,d2+β14,yDs,y,d3.

For convenience, this equation can be written in matrix notation as (7) Yy=ByXs,d,y

where X is a data matrix with l columns (l = 4; the number of year-level predictors) with the first column a vector of 1’s for the intercept and By is the y × l matrix of year-level regression coefficients. Priors for the mean were non-informative, with (8) By∼MVNMl,Σ

where Ml = (μα, μβ12, μβ13, μβ14) is a vector of length l, representing the mean of the distribution of intercept and slopes. The l × l covariance matrix is represented by Σ where the variance of each regression coefficient is on the diagonal and the covariance on the off-diagonals. The hyperprior for the means were non-informative with μα, = 0 and μβ12, μβ13, μβ14 ∼ N(0, 100). Standard deviation priors were also non-informative and were drawn from an inv-Wishart distribution: (9) Σ∼inv−wishdiagll+1.

Parameter estimation

We used the program JAGS (http://mcmc-jags.sourceforge.net) to code the model and to draw posterior samples of the parameters (see Supplemental Information for JAGS code). We called JAGS from R (3.1.2) using the package ‘rjags’ (V 3-14). We ran three chains with 1,000 burn-in and 2,500 evaluation iterations. Chains were thinned to keep every fifth iteration. We checked convergence using the ‘potential scale reduction factor’ (Brooks & Gelman, 1998) from the ‘coda’ package in R (Plummer et al., 2006) and also assessed chains visually.

Model assessment

Goodness of fit and prediction

We assessed goodness of fit in two ways. First, we compared observed and predicted values for the complete dataset. Second, we ran a series of cross validation tests where we randomly left out a portion of the water temperature data, estimated parameters with the remaining (training) data and compared predictions of the left out (testing) data to original values. This involved leave-p-out cross-validation where we randomly left out a proportion (p) of the data, where p = 0, 0.05, 0.1, 0.2, 0.3, 0.4, 0.5, 0.6, 0.7, 0.8. We ran 10 replicates for each value of p. For each condition, we calculated the root mean square error (RMSE) of the residuals for the training and the test data sets.

Missing data

We also ran a series of tests to ask how the quantity, timing, and location of missing data influenced model performance (estimation and prediction). These tests can be used to help understand performance and to help design monitoring strategies. This set of analyses differed from the leave-p-out cross-validation (above) because data were not left out randomly. Rather, consecutive days of data were left out, either within a year or across streams, reflecting the character of missing field data.

Quantity: To evaluate how increasing the number of sampling days within a year affects estimation and prediction, we left out increasing numbers of days on either side of the median sampling date for each stream and year combination. Specifically, we started with complete data and then conducted nine sets of runs where we left out data 15⋅d days from the beginning and 15⋅d days from the end of each time series (where d = 1–9), generating shorter time series by 30 days for each scenario.

When: We assessed how changing the timing of missing data affected predictions by shifting the window of available data from the beginning to the end of the synchronized period. To do this, we left out data for all but 30 consecutive days at a time for 13 non-overlapping scenarios with scenario one starting at day of year 70 and scenario 13 starting at day of year 310.

Where: To evaluate how well the model predicted stream temperatures when data were missing from one or more streams, we ran the above analyses leaving out data yearly from all streams or just the West Brook. For years with data just from the West Brook (1999–2002), we removed all data a year at a time. For years with data from the tributaries and the West Brook (2003–2013), we either removed all the data for each year (all four streams) or just the data from the West Brook for each year. Removing all the data for a given year tests how well the model predictions work when there are no data for the year (but there are data for other years), while removing data for just the West Brook tests how well predictions work when data are missing for a stream (but there are data for other streams and years).

For all tests, we compared RMSE of the residuals for the test (left out) data to the RMSE of the residuals of the full training set (base case).

Results

Descriptive statistics

Evaluation of the descriptive statistics suggested that water temperatures were similar for OL and IL and for WB and OS and that the streams appear to be warming over the duration of the study. Correlations of daily water temperatures among the four sites were all between 0.96 and 0.97, except for the correlation between OL and IL (0.99). Patterns in the cumulative water temperature residuals were generally similar for WB and OS, with cooler years in the beginning of the time series and warmer years later (Fig. 4). OS demonstrated the warmest temperatures, especially in 2010–2012. Patterns were remarkably similar between OL and IL, also demonstrating generally cooler temperatures earlier in the data time series (Fig. 4). Monthly distributions of water temperature were highly variable across years and streams (Fig. S1).

Figure 4 Cumulative residuals from the spline in Fig. 2 for each site and year combination.

Curves on or near the horizontal line indicate ‘typical’ years whereas curves above the line indicate warm years and below the line indicate cool years.

Parameter estimation

Potential scale reduction factor (R-hat) values for all parameters were less than 1.01, indicating good convergence (Brooks & Gelman, 1998). Parameter estimates gave an overall mean of 15.1, with strong air temperature effects (1.52 unlagged, 0.20 lagged 1 day, 0.15 lagged 2 days), a positive effect of stream flow (0.36), and strong site differences (OL = − 0.50, OS = 0.59, IL = − 0.54) (Fig. 5 and Table S1). The autoregressive mean equaled 0.79 and there was little variation in the autoregressive terms among sites (Fig. 5). The estimate for residual model error from Eq. (1) was 0.77.

Figure 5 Parameter estimates from the stream temperature model.

‘T’ stands for temperature, ‘F’ stands for flow, and OL, OS, IL are streams. The first 12 rows represent the βx in Eq. (5). ‘ar1[x]’ are the δx from Eq. (3), and the ‘Linear’, ‘Quadratic’ and Cubic’ parameters are the μYx from Eq. (7). ‘sigma’ is the residual standard deviation from Eq. (1).

Model assessment: goodness of fit and prediction

Using the full dataset, predicted values were very similar to observed values (Fig. S2). The slope of the relationship was 0.99 (s.e. = 0.0064) with an intercept of 0.15 (s.e. = 0.089) and an R2 of 0.98. Overall RMSE was 0.59 ± 0.09 (Table 3). For the cross-validation tests where we randomly left out 30% of the data, the RMSE increased to 0.69 ± 0.003 for the training data and to 0.86 ± 0.010 for the test data (Table 3).

Table 3 Root mean square error (RMSE, °C) for various scenarios described in the text.

The scenarios involved a training dataset and a test dataset (data left out).

Scenario	RMSE train	Test streams	RMSE test	RMSE difference	
All data	0.59 ± 0.09	–	–	–	
30% data randomly left out	0.69 ± 0.003	All	0.86 ± 0.010	0.17	
For each year, West Brook left out	0.59 ± 0.09	West brook	1.07 ± 0.26	0.48	
For each year, all streams left out	0.59 ± 0.09	All	1.16 ± 0.35	0.57	

Across a broader range of data randomly left out (0–0.8), the RMSE for the test data increased approximately linearly with a 0.025 increase in RMSE for each 0.1 increase in proportion of data left out (r2 = 0.98; Fig. S3). RMSE for the training data set was largely insensitive to the proportion of data left out and had a mean value of 0.86 (s.d. = 0.016, Fig. S3).

Break point trends

Break points appear to be getting later in the year in the fall and earlier in the year in the spring (Fig. 6). In the fall, ΔAIC values for the linear models were all within two so we selected the simplest model (year only). In the spring, the ΔAIC value for the simplest model was 5.2 so we also selected the simplest model (year only). Both seasons showed significant changes in breakpoints over years (estimate = 1.33, F(1, 40) = 4.68, p-value = 0.036 fall; estimate = − 1.61, F(1, 40) = 9.13, p-value = 0.0045 spring), but year explained only 10% (fall) or 19% (spring) of the variation in the relationship. The parameter estimates indicated that breakpoints are 16 days earlier in the spring and 13 days later per decade in the fall, generating an estimated widening of the breakpoint window of 29 days per decade.

Figure 6 Fall and spring breakpoints across years for the four streams.

Trends in cubic functions

Predicted mean water temperatures based on the cubic function (Eq. (7)) varied among years (Fig. S4), mirroring the general trend in the raw data (Fig. 2). Yearly maximum water temperature (white dots in Fig. S4) increased over the course of the study (F = 5.34, df = 1, 13, p-value = 0.037, R2 = 0.24), with an estimated decadal increase of 0.63 °C (Fig. 7, above). In contrast, the day of year of the temperature maximum did not change over the course of the study (F = 0.030, df = 1, 13, p-value = 0.86) (Fig. 7, below).

Figure 7 Predicted maximum temperature for each year (y axis value of dot in Fig. S4) and predicted day of the maximum temperature (x axis value of dot in Fig. S4).

Shadows represent 95% confidence bands.

Missing data: quantity

Adding more data to either side of the median date improved predictions of the test data (filled circles in Fig. 8). The slope of the regression (−0.46, s.e. = 0.035) indicated that a 10% increase in data resulted in a reduction in RMSE of 0.046.

Figure 8 Root mean square error (RMSE) difference from the base case (all data included) for the WB for the cross-validation analyses changing the proportion of days included in estimation.

Estimation data included either no data from any of the streams for each year (triangles, dashed line) or data from the three other streams but no data for the WB for each year (circles, solid line).

Missing data: when

The timing of data availability had a threshold effect on RMSE, with relatively high and variable RMSE before day 160 and consistent lower RMSE after day 160 (triangles in Fig. 9).

Figure 9 Root mean square error (RMSE) difference from the base case (all data included) for the WB for the cross-validation analyses changing the starting day of a non-overlapping 30-d moving window.

Estimation data included either no data from any of the streams for each year (triangles) or data from the three other streams but no data for the WB for each year (circles, solid line).

Missing data: where

Compared to the base case (all data included), leaving data out of the estimation one year at a time resulted in a mean increase in RMSE of 0.48 °C when just the WB data were left out and a mean increase of 0.57 °C when data for all four streams were left out (Table 3).

As the amount of data was increased on either side of the median date, RMSE increased less from the base case when data were available from the other three streams than when data were not available for any of the streams. The slope of the relationship between the proportion of days included in the training data and the difference in mean RMSE was −0.46 when just the WB data were left out and the slope was −0.63 (s.e. = 0.20) when data from all streams were left out (Fig. 8). The slopes suggest either a 0.046 or a 0.063 decrease in RMSE with a 10% increase in days included in the estimation.

When data were available for only 30 days, but the 30-day window of availability varied across the year, the presence of data from the other streams eliminated the variability in RMSE across scenarios (compare circles to triangles in Fig. 9). The resulting increase in RMSE was about 0.38 across 30-day window scenarios when data were present from other streams.

Discussion

We present a statistical model that accounts for many issues that can make stream temperature estimation difficult. Our model limits analysis to days when air and water temperature are synchronized, accommodates hysteresis, incorporates time lags, can deal with missing data and autocorrelation and can include external drivers. The result is quite low bias with complete data (RMSE = 0.59 °C), and bias remains low (RMSE < 1 °C) when data from streams or years are missing.

A key feature of our model is a flexible way to identify the portion of days spring-to-fall when daily stream and air temperatures are synchronized. The air-water temperature relationship breaks down during the winter, primarily, due to phase change thermodynamics, insulating ice cover, snow melt, and other physical processes. Previous researchers have omitted modeling winter temperatures or focused solely on summer temperatures (e.g., Kanno, Vokoun & Letcher, 2014; DeWeber & Wagner, 2014; Snyder, Hitt & Young, 2015). However, defining the “winter” period that causes deviations in the air-water relationship depends on the conditions in a specific year and location; therefore, just excluding the winter months based on calendar dates (21 December–20 March in the northern hemisphere) is an imprecise cutoff with the potential to bias the model and the resulting inference. For example, just as the amount of snow and duration of ice cover differs at 40°and 45°latitude, the physical properties that affect the air-water relationship vary annually and from one location to another depending on the exact landscape characteristics of the site, even when compared to nearby locations (Lisi et al., 2015). Additionally, taking the opposite approach and limiting analyses to the summer period excludes large amounts of data and prevents inference during other times of the year, which are important in biological and biogeochemical processes. Our method of calculating the period of the year where the air-water relationship is synchronized alleviates these issues of arbitrarily defining the winter period while maximizing the amount of data available for modeling linear effects of air on stream temperature.

Modeling the synchronized period of the year also provides additional information about the spring and fall breakpoints and the duration between them. Despite considerable random annual variation, we found that air-water relationships were getting synchronized earlier in the spring and remaining synchronized later in the fall. This resulted in a 29 days per decade expansion in the synchronized period of the year or 44 days over the 15-year study period. This has implications for the growing season (e.g., algal growth, primary productivity, nutrient cycling), which affects invertebrate (Ward & Stanford, 1982) and vertebrate growth and development (Neuheimer & Taggart, 2007; Venturelli et al., 2010). Growing seasons worldwide have been expanding about 10–20 days over the last few decades (Linderholm, 2006), which is slower than the expansion in the synchronized period we observed. The relationship between plant-based growing season estimates and the width of the synchronized period is currently unknown, but further model development could help identify links between the width of the synchronized period and other growing-season metrics.

Hysteresis is another challenge when modeling stream temperature (Webb & Nobilis, 1997). We allowed for the potential differences in seasonal warming and cooling with a cubic effect of day of the year on water temperature (Figs. 2 and S4). This can be understood as the average expected water temperature on any day of the year during the synchronized period. Then the effects of air temperature, flow, and site can be thought of as moving the water temperature away from this mean expectation. We also allow this cubic effect to vary randomly by year. This has two major benefits. First, it allows the idiosyncratic seasonal temperature patterns to vary annually (Fig. S4). Otherwise it would be very difficult to have a parametric model describing the effects of a warm, wet spring followed by a cold summer or three moderately cool weeks followed by one extremely hot week in the autumn. The second benefit is, by having a random year effect, the pattern of hysteresis is variable and can be well-described when sufficient data are available, while in years with little data the predictions move towards the mean across years. This borrowing effect allows for good predictions even in years with minimal data. An alternative to the parametric cubic function is a non-parametric smoothed function, but it can be challenging to estimate hierarchical effects for smoothed functions. Li et al. (2014) present a stream temperature model with time-varying smoothed functions which allows parameter estimates to vary over time. The time varying coefficients can account for variation in the air-water temperature relationship that is not included in the model. RMSE estimates (∼1 °C) from the time-varying coefficient model are low and similar to the estimate from our model (0.59 °C).

Using the cubic function also provides information on the smoothed annual peak temperature and the date of the peak water temperature. We estimated that the peak temperature increased at a rate of 0.63 °C per decade, or 0.94 °C over the course of 15 years. The stream temperature warming rate is within the range of rates identified in rivers and streams across the US (0.07–0.77 °C per decade, Kaushal et al., 2010), three-fold faster than the rate identified using simple linear models in the Chesapeake Bay watershed (0.28 °C per decade, Rice & Jastram, 2015) and twice as fast as warming of lake surface water in the summer (0.34 °C per decade, O’Reilly et al., 2015). In contrast to the peak temperature, the day of the year that the peak temperature was reached did not change during the study. This decoupling between the value of the peak and the day of the peak suggests that increased peak temperatures are not a result of a change in the timing of maximum temperatures, but rather are driven primarily by increased air temperatures.

Our hierarchical approach to modeling handles years and sites with varying amounts of incomplete data. A hierarchical model can accommodate missing data for one year and site by ‘borrowing’ information from other years and sites (Bolker et al., 2009). We evaluated how missing data influenced prediction bias across years and sites. When data for a single year and all sites were left out, bias (increase in RMSE) was higher (+0.57) than when just the West Brook data were left out (+0.48), demonstrating how data from nearby streams can inform estimates. It will be important in the future to identify the strength of the spatial decay function to understand how close sites (on the network) should be to allow effective information sharing.

We also evaluated how missing data within years influenced predictions. First, we added data from the middle of the year in both directions and found that a 10% increase in data resulted in approximately a 10% improvement in RMSE. Clearly, more data during the synchronized period will provide better predictions, but predictions can still be reasonable with limited data during the year. This may be especially true when data from more nearby streams are available, as stream temperature monitoring becomes increasingly common. Second, we evaluated how data availability during the year affected predictions by retaining 30 days of data and shifting the window of availability across the year. When data were available from the other three streams, WB predictions with missing data were insensitive to the timing of available data (consistent 0.38 increase in RMSE). However, when data from the other three streams were not available, predictions were poorer when data were only available early in the year compared to late in the year. When data are available from nearby streams, the local data can help define the annual cubic pattern in the model, but when they are not available the higher variability in daily stream temperature in the spring compared to the autumn likely results in some years with cubic patterns that are a poor fit to autumn stream temperatures.

We used a simple autoregressive term to model the temporal autocorrelation in the residuals. This is critical in a regression-based daily temperature model because the error at time step i is likely to be correlated with the error at time i + 1 due to some small temporal variation not accounted for by the regression parameters. Any autocorrelation or patterning in the residuals violates the assumptions of a linear regression model. This is a classic problem in time series analysis (Shumway & Stoffer, 2006). In our model, the AR1 term adequately corrected for temporal autocorrelation such that the resulting residuals displayed homogeneity and were normally distributed. No additional lagging or moving average was needed in this case, but it would be easy to add these additional ARIMA parameters to the model if necessary. The estimate of 0.79 ± 0.05 (mean ± s.d., Table S1) for the autoregressive term indicates strong effect of the previous day’s residual on stream temperature.

Air temperature can be used as the primary variable predicting water temperature in small streams. However, additional factors can influence water temperature directly or affect the air-water temperature relationship (Caissie, 2006; Sun et al., 2014). We found that the effect of air temperature was reduced as stream flow increased (significant negative coefficient; Table S1). This corresponds to our expectations because a larger volume of water will require more energy to heat and at high flows the streams are generally deeper resulting in a lower relative surface area in contact with the air. Additionally, higher flow is often a result of surface and ground water inputs originating over the previous days and weeks and therefore less influenced by the current air temperature. Here, flow was our only external variable. Our model structure can easily accommodate additional factors such as forest cover, agriculture, impervious surfaces, impoundments, and ground water when these data are available and vary over the streams of interest. Incorporating groundwater data at broad spatial scales, however, remains a substantial challenge due to a lack of a consistent, accurate data layer. The model could also easily be extended to model daily minimum (Hughes, Subba Rao & Subba Rao, 2007) or maximum (Caissie, El-Jabi & Satish, 2001; Li et al., 2014) stream temperature in addition to the daily mean modeled here.

Local variation of environmental drivers at very small spatial scales can have a strong influence on stream temperatures. For example, ground water input can moderate air temperature effects in the summer and winter (Poole & Berman, 2001; Kanno, Vokoun & Letcher, 2014; Westhoff & Paukert, 2014; Snyder, Hitt & Young, 2015). We did not model groundwater effects because we lack information on the spatial distribution of groundwater inputs in our small system, but we did observe marked differences in water temperature across streams (Fig. 4). Water temperatures in the WB and OS were considerably warmer than temperatures in OL and IL. Stream-specific intercepts reflect the raw stream temperature data, with a range of 1 °C across streams (stream-specific parameters in Table S1). The most likely explanation for the temperature differences is the presence of upstream impoundments (Webb & Walling, 1996; Dripps & Granger, 2013); the warmer streams have either an upstream reservoir with a surface release (WB) or a beaver impoundment (OS). The cooler streams do not have any impounded water. Even small temperature differences among streams can have important consequences for production and phenology of stream biota (Quinn et al., 1994; Miller et al., 2011; Wheeler et al., 2014; Letcher et al., 2015), reinforcing the value of statistically robust stream temperature models.

By accounting for many of the issues that make stream temperature estimation difficult, our stream temperature model provides robust estimates with low error. Most current stream temperature models do not address all of these issues and generally report higher error rates, especially models of daily stream temperature. One reason error rates of our model are low is that we limit analysis to the synchronized period, but this has the added benefit of generating data for the beginning and end of the synchronized period which can be very useful for evaluating shifting stream phenology. Our model can also accommodate missing data which, unfortunately, is common in streams as temperature logger availability limits data to incomplete spatial coverage and often incomplete temporal coverage within a year. The structure of our model is flexible enough that a data time series even as short as 10 days can contribute important information.

Supplemental Information

Supplemental Information 1 Supplemental material

Click here for additional data file.

Supplemental Information 2 Supplemental information

Click here for additional data file.

Table S1 Parameter r-hat values, means, and credible intervals

Click here for additional data file.

Figure S1 Violin plots of raw temperature data

Violin plots of monthly daily mean water temperatures across years (panels) and sites. Filled circles show mean for each month, open circles show means for each month for each stream, and lines show the mean for the month, stream, year combination.

Click here for additional data file.

Figure S2 Observed vs. predicted

Observed vs. predicted daily water temperatures for combinations of site and year. The line in each panel is the 1:1 line.

Click here for additional data file.

Figure S3 Test of missing data

Root mean square error of the test data set (triangles) and training data set (circles) for 10 replicates of runs with increasing proportions of data left out of the training set.

Click here for additional data file.

Figure S4 Predicted cubic temperature curves

Predicted cubic curves for each year against day of year (Eq. (7)). The white dots signify the yearly maximum water temperature and the day of maximum water temperature.

Click here for additional data file.

Additional Information and Declarations

Competing Interests

Author Contributions

Data Availability

Benjamin H. Letcher is an Academic Editor for PeerJ.

Benjamin H. Letcher conceived and designed the experiments, performed the experiments, analyzed the data, contributed reagents/materials/analysis tools, wrote the paper, prepared figures and/or tables, reviewed drafts of the paper.

Daniel J. Hocking analyzed the data, contributed reagents/materials/analysis tools, wrote the paper, reviewed drafts of the paper.

Kyle O’Neil reviewed drafts of the paper, and analyzed and collected data.

Andrew R. Whiteley and Matthew J. O’Donnell performed the experiments, reviewed drafts of the paper.

Keith H. Nislow conceived and designed the experiments, performed the experiments, reviewed drafts of the paper.

The following information was supplied regarding data availability:

Stream temperature data are available in the temperature database at:

ecosheds.org.

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
