# Peer review of "A hierarchical model of daily stream temperature using air-water temperature synchronization, autocorrelation, and time lags"

_PeerJ, doi:10.7717/peerj.1727_

## Round 0.1 · original submission · Major Revisions

· Academic Editor

Major Revisions

Both reviewers saw significant merit in the paper, as did I. I agree that the statistical approach is robust for the data sets and particular streams that were the basis of the study. How universal it is remains to be demonstrated. Both reviewers and I think that the discussion and conclusions go too far beyond the data. Because of that, the paper is way too long. It needs to focus on the characteristics of the particular streams that were studied and the advances in the statistical design. Be straightforward about the well-known aspects of stream ecology that can add variance to correlation of air-water temperatures, such as upstream impoundment and ground-surface water interactions. Incorporation of such "landscape drivers" just is not "an easy add on". You do not robustly test landscape drivers and indeed figuring out how to do that in any kind of organized classification or ordination basis--whether regionally or globally-- remains a large and formidable challenge. It certainly cannot be tackled if we continue to black box groundwater-surface water interactions as you have done. This underscores that your study is grounded in the characteristics of the streams where data were gathered (i.e., you would have different results comparing outcomes from your streams, say, to floodplain rivers). Your big contribution is the modeling approach, so focus on that.

For the paper to be acceptable in PeerJ, you must shorten and focus the paper specifically around your data sets and statistical approach for your study streams. The introduction can be shortened, perhaps by 50%, but your literature citations all are appropriate. Please carefully consider and rebut all of the reviewer comments.

·

Basic reporting

The writing is very good. The introduction provides a comprehensive treatment of the literature and mechanisms driving stream temperature: thorough yet concise. See 18 Dec 2015 issue of Science pg. 1449 and reference therein for recent results on warming in lakes and potential mechanisms (important in the context of this study, since the streams that showed considerable temperature deviations had a significant % of lake area in their catchments). Occasional problems were noted in labeling the figures (e.g. site codes differ from those defined in the text). Some errors in figure citations (Table A1 and Figure A1?).The methods section is more or less clear although some of the terminology that might be unfamiliar to readers with less experience in time series statistics might be better defined or at least referenced. In particular the term “synchronized” should be defined to apply to the annual time scale (not to be confused with phase and amplitude synchronization driven by shorter term hydrologic events). Not clear whether raw temperature data has been archived, or whether this level of open access is required by the journal (a problem specific to very large datasets generated by data loggers, and their metadata, including calibration and sensor maintenance and replacement).
A few specific observations noted in passing:
Line 461 change “charge” to “change
Line 176: Should 2013 should actually be 2008, to correspond with following sentence?
Recommend labeling of B coefficients in figure 5 and table S1 (cited in the text as A1) to correspond with equation 5 (otherwise the reader has to continuously cross reference).
Xu reference mis-cited

Experimental design

It should go without saying that sensor replacement and calibration procedures would have an important effect on results (see comments above on data archiving). Sensor drift could easily account for some of the warming trends reported here, especially over multi-decade deployment times (but would have less effect on the timespan between break points). Given that the data logger model not mentioned (TidBit?), and that the cited sampling frequency and resolution does not correspond with that reported in previous studies at the same site by the same authors: calibration and maintenance procedures need to be addressed.
Using surrogates for air temperature and streamflow may be justifiable for this site (and certainly the model performance suggest this), but in the interest in characterizing model robustness it would be important to know distance relative to nearest sites used to validate air temperature (Daymet model) and streamflow (Mill River). For application to other systems the models may not perform as well, depending on these and other factors. It is presumed that daily average air temperature from Daymet was used?
In this context of site selection discussed above, the authors should clarify in the site description that this in an area of low to moderate relief. A table of elevation, watershed area and perhaps %lake area for each basin would be more useful than the general values for elevation range and area for the WB basin), where assumptions on 1km modeled air temperature are valid, but not for areas with higher relief (hence the model is locally robust, but will likely encounter greater challenges for other sites).
Calculation of the flat period and breakpoints based on 150 day CI may be unduly sensitive to intra-annual variability in temperature variability, especially the amplitude and duration of diel cycling (that is, extending the C.I. would also extend the period defined here as synchronized, not by an increased period of synchronization but by allowing for a wider tolerance for defining it). Why not calculate rate of change or inflection point for TempIndex and establish a common threshold for all sites or site specific threshold?
Table 1 data set is notably incomplete (understandably so, see comments below). It is suggested that the authors include information on whether data gaps are from the winter time period (excluded from analysis), or are there significant holes in the periods used for analysis (synchronization period)?

Validity of the findings

The TempIndex metric used to determine seasonal synchronization is basically an indicator of when the difference between air and stream temp is relatively small in comparison to stream temp (large denominator producing values <<1). Hence the robustness of the metric depends on latitude and elevation (with tropical and mountain watersheds probably not producing as clear an index) and also watershed size (decoupling between downstream observation point and upstream watershed drivers). These should be considered in the context of self-described model robustness.

Missing data and incomplete time series for temperature loggers is a common problem, despite their convenience it is often impossible to generate perfect time series, a phenomenon that is perhaps deserving a corollary of Murphys Law (Michelle’s Law?). In this sense the practical applications of filling voids in data and analysis of effects of data gaps is a very useful and important result of this work. However it remains to be tested for robustness in a wide range of systems: I doubt that it would work well in even closely spaced mountain snowmelt streams (variable aspects within the watershed and variable timing of snowpack ripening).
The very incomplete time series for sites used in this study, and lack of clarity as to the degree of effect upon intervals needed for analysis, results in some ambiguity when interpreting the results, especially when drawing conclusions about warming. Is it a model, or is it data? Depending on the amount of missing data and model performance for any given site, the answer is somewhere in between (depending on site and year), the results are part model and part empirical observations. Any given conclusion might be better qualified by including %data substitution (modeled) and RMSE specific for that result.
In the same context, some general advice on reporting the outcomes from this work: many of the conclusions such as breakpoints (lines 422-424), warming rate (lines 455-457) and peak temperature (lines 4161-464) would be far more conclusive based on complete temperature records. The authors avaluate model sensitivity on a general level, but it is not clear how much modeled data contributes specifically to each result(data)/outcome(model) for respective sites. Also, reporting per year warming rates (3 decimal places) belies the reported precision of the temperature loggers (1 decimal place), and also may not represent inter-annual variation (including ENSO and decadal cycles). Perhaps it would be better to report rate of change per decade, as is often done. This may seem trivial, but would be more consistent in the context of the very precise terminology and manner that predictions are parameterized in international climate change reporting (e.g. IPCC).
Some discussion might be useful on extending the metrics to include those other than mean values (why are we overly fixated on means?). Degree days may be more relevant in terms of growth, bioenergetics and life cycles (see Stanford reference cited in the intro), minimum temperature in terms of biogeographical distribution (reference dating back to the late 1800s by Dana, reprinted in Foundations of Biogeography) and maximum in terms thermal tolerance (e.g. trout). The last may be disproportionally affected by climate change relative to changes in mean water or air temperature.

In general, this is excellent work and clearly fills a void on the analysis of stream temperature time series, and especially regarding the common problem of data gaps. On the other hand, it might be mentioned that at some point the cure becomes worse than the disease: maintaining a calibrated gap free sensor network is always a challenge, but may be far less complicated than the analysis needed to fill the gaps post hoc, and the resulting increase in uncertainty.

·

Basic reporting

While I think there are some useful and new approaches and findings presented in the MS, I think it suffers from a lack of focus. It could benefit from revision to better clarify and more narrowly define the scope and intent of the paper, especially with rewriting of sections of the Introduction, Methods, and address a few points accordingly in the Discussion and Abstract. In my comments that follow I'll try to specifically identify the elements I found new and useful, one or two claims I found to be too much of a stretch for this paper, and make some suggestions for remedy. I do not have sufficient expertise and time to evaluate the mathematical modeling and every aspect of the statistical methods, though I did not have any problem following the logic sequence and outline of the analytic and empirical bases for the results and conclusions. These relations are well-represented in the Figures.

Experimental design

The experimental design of this study is far from ideal. However, it has the considerable virtue of being highly realistic. That is, shortcomings of the design and data streams are familiar and well representative of shortfalls that are commonly, in fact almost universally, seen in these kinds of field data. The spatial and temporal dimensionality of the data are very typical of what researchers and resources managers commonly face in watershed-specific field studies

I think the Title, Abstract, Introduction, Methods, and Discussion should be revised to reflect a more narrow focus on developing and demonstrating useful analytic methods to evaluate and detect time trends in stream temperature data. That is, in my opinion the design and scope of the study are not in fact sufficient to justify a universally applicable simulation model for modeling stream temperatures across multiple scales or space and time, as the MS seems to imply. There are few or no data or methods here to directly inform, evaluate, or predict landscape influences on hydrologic mechanisms and dynamics, including groundwater-surface water interactions, despite that the text mentions this as a recognized necessity for a general model.

Nevertheless this MS provides a very useful example of some new conceptual and analytic approaches for making the best use of available stream temperature data for one particular case study. These approaches and methods are certainly applicable (though to varying degrees, yet to be ascertained) to other cases, regardless of whether the "model" as a whole developed for these specific streams is immediately transferable to other streams, stream types, and regions (I would suggest there is good reason to believe it is NOT--including, e.g., the authors' comments on stream size effects that are outside the scope of these data).

In the methods section one item tripped me up-- that is, the unsubstantiated and vaguely worded statement in lines 180-182 that stream water temperatures are thermally controlled by "energy sources over a larger area." While the sense of weather and climate effects this might be true to a degree, it certainly has been shown that localized energy in the form of microclimate controls (e.g., reach-scale shade, see Oregon work by J. Groom and colleagues, earlier work also in Oregon Hostetler and Brown also many others) and groundwater and hyporheic influences (numerous papers pertain here) also exert strong controls over stream thermal regimes. I don't think the current state of knowledge necessarily supports a cavalier assumption that a regional air temperature record will be a better predictor of stream temperature than a site-specific measure of air temperature (especially across the multiple scales of space, time, and stream size that are applicable)--nor does it appear to me to be a question that is actually tested in this paper. It remains an open question that could be better tested, however, if future investigations consider employing some of the analytic techniques demonstrated here. In fact, in terms of design, the poverty of the air temperature record is the most concerning limiting factor in the present study. But, that is also an all-too-typical limitation in most real world data.

The quantitative approach to identifying seasonal qualitative breakpoints in thermal relations between air and water temperature, the methods for testing the effect of data omissions and gaps (and running these forward, for identifying design and sampling needs relative to desired precision and acceptable error), and the methods for detecting interannual time trends in stream temperature and thermal regime strike me as new and useful contributions, or at least substantive refinements of previous methods, and I think these should be the express focus of this paper.

Validity of the findings

The present Results section, though it could perhaps be sharpened a bit, actually well reflects my suggestions above as to what the primary contributions of this paper are. The results appear to follow logically from the data and analyses, though their logical importance is somewhat lost in the current formulation of the paper. Sharpening of the Intro and Methods as suggested above what provide the context the reader needs to generally infer why these specific Results are important, while leaving more specific explications to the Discussion.

For me the Discussion gets off to a bad start in its first paragraph. I found this summary of intent and results to be hard to follow and therefore also unconvincing, but the claim in Lines 397-398 that it is "straightforward to extend the model to a broader spatial scale..." seems to stray well off the reservation. The design is not adequate to test this question of extrapolation or regional robustness of the simulation modeling that was developed and tested solely on this limited set of small study streams. It's also just not necessary to make this claim, which will strike many informed readers as naive, at least as it is currently worded.

However, there follows some good stuff. Here are the lines (roughly listed) that struck me as reporting important material and successfully pointing out how the findings relate to and improve on prior work: Lines 399-405, 406-409, 415-418, 421-424, 454-459, 460-461, 486-490, 546-556.

The intervening paragraphs that are not touched by the above-listed sections seemed to me to wander some and consist mostly of restatement of general background material that was presented more concisely as essential premises early in the paper, and are not here effectively tied to the specific results of this study. I'm not saying some of this isn't potentially important material, but to make its importance clear will take a more cogent relation to your specific results.

Additional comments

I found the analytic ideas and content for better linking air and water temperatures interesting and useful, and I hope the authors will pursue revisions to focus on those aspects. We all have a long way to go on the larger questions of relating empirical air and water temperature correlations to characteristic hydrologic mechanisms and processes, in a way that is scaleable and can inform and underpin a truly robust and universal model and classification.

---

## Round 0.2 · accepted · Accept

· Academic Editor

Accept

I think you responded forthrightly to the comments of the reviewers. You did not reduce the introduction as I requested but I understand and now agree with the need to provide a complete background review to support the approach of the study you conducted. I am satisfied that this paper is ready for publication in PeerJ.